# Coordination of cell envelope biology by *Escherichia coli* MarA protein potentiates intrinsic antibiotic resistance

**Alexandra E. Trigg** **, Prateek Sharma, David C. Grainger** *

School of Biosciences, University if Birmingham, Edgbaston, Birmingham, England

* d.grainger@bham.ac.uk

## Abstract

The multiple antibiotic resistance activator (MarA) protein is a transcription factor implicated in control of intrinsic antibiotic resistance in enteric bacterial pathogens. In this work, we screened the *Escherichia coli* genome computationally for MarA binding sites. By incorporating global maps of transcription initiation, and clustering predicted targets according to gene function, we were able to avoid widespread misidentification of MarA sites, which has hindered prior studies. Subsequent genetic and biochemical analyses identified direct activation of genes for lipopolysaccharide (LPS) biosynthesis and repression of a cell wall remodelling endopeptidase. Rewiring of the MarA regulon, by mutating subsets of MarA binding sites, reveals synergistic interactions between regulatory targets of MarA. Specifically, we show that uncoupling LPS production, or cell wall remodelling, from regulation by MarA, renders cells hypersensitive to mutations altering lipid trafficking by the MlaFEDCB system. Together, our findings demonstrate how MarA co-regulates different aspects of cell envelope biology to maximise antibiotic resistance.

## Author summary

Understanding the processes that give rise to antibiotic resistance is a major societal challenge. In our paper, we have studied antibiotic resistance in the bacterium *Escherichia coli*. Specifically, we sought to understand how the microbe can increase its resistance to antimicrobial substances by using different sets of genes. We have discovered that, by altering which genes are in use, *E. coli* can change the properties of the barriers that protect the cell's internal components from the outside environment. In this way, the organism can reduce entry of antibiotics and so avoid their harmful effects.

## Introduction

The *Escherichia coli* multiple antibiotic resistance (*mar*) locus is a chromosomal determinant, common to many enteric γ-proteobacteria, important for cross-resistance to tetracyclines, quinolones, and β-lactams [1–5]. The phenotype is mediated by a single operon, known as

**Data availability statement:** All data are available via the figures and Supplementary information

**Funding:** This work was supported by a BBSRC studentship to AET and a Wellcome Trust Investigator award to DCG. The funders had no role in study design, data collection and analysis, decision to publish, or preparation of the manuscript.

**Competing interests:** The authors have declared that no competing interests exist.

*marRAB*, within which the first two genes are key. Briefly, *marR* encodes a transcriptional repressor that autoregulates the locus, but has no other target genes [6–8]. Conversely, *marA* encodes a global regulator of transcription with many targets [7,9–15]. Changes in gene expression controlled by MarA give rise to multiple antibiotic resistance [3,5,16]. In wild type bacteria, MarA is produced if MarR releases its operators upstream of *marRAB*. This can be triggered by phenolic compounds [17] and copper dependent disulphide bond formation between MarR molecules [18]. Additionally, clinical *E. coli* isolates, resistant to antibiotics, may encode mutations that inactivate MarR or destroy its cognate DNA site [10,6]. Such mutants are readily selected in the laboratory by treating *E. coli* cells with antibiotics [1,10]. Importantly, since MarA lacks its own signal sensing domain, mutations that circumvent repression by MarR cause constitutive induction of the MarA regulon. Like all AraC-family transcription factors, MarA binds a non-palindromic DNA target [19]. This DNA element, known as the "marbox", is conserved in many enteric bacteria and can also be recognised by the closely related MarA-like transcription factors, SoxS and Rob, in *E. coli* [20]. SoxS is expressed in response to oxidative stress sensed by the upstream regulator SoxR [21]. Unlike MarA and SoxS, Rob directly detects environmental signals and is induced by dipyridyl [22]. A factor called RamA, not present in *E. coli*, also binds the marbox in other Enterobacteriaceae [9,23–25]. Promoters activated by MarA-like regulators can be divided into two classes. At class I promoters, the marbox is in the reverse orientation and upstream of the -35 element. Class II promoters have a forward orientation marbox overlapping the -35 hexamer [26].

The ability of MarA, and related factors, to drive antibiotic resistance, has long been attributed to increased expression of drug efflux pumps and inhibition of porin expression [27,28]. However, these processes are unable to explain all drug resistance phenotypes. For example, constitutive MarA expression is still able to stimulate minocycline resistance in cells lacking *tolC* or *acrAB* [15]. Hence, numerous studies sought to identify direct regulatory targets of MarA in enteric bacteria. Early attempts relied on searching for occurrences of the marbox in genome sequences [29,30]. However, this approach is troublesome because MarA binding sites have a comparatively low information content [7,9]. Consequently, around 10,000 copies of the marbox can be identified across the *E. coli* chromosome [29,30]. Only a small fraction are likely to be functional for gene regulation [29,30]. The advent of whole transcriptome analyses allowed determination of genes differentially regulated in the presence and absence of MarA, or closely related regulators, such as SoxS and Rob [31,32]. These studies identified some direct regulatory targets but data interpretation is difficult for two reasons [13]. First, the majority of gene expression changes are indirect consequences of pleiotropy [9,13]. Second, because MarA, SoxS, Rob and RamA are so similar, and share many binding sites, loss of one regulator can be compensated for by the other factors [9].

In the last ten years, chromatin immunoprecipitation (ChIP) coupled with deep sequencing (ChIP-seq) has emerged as a useful tool for mapping the direct regulatory targets of transcription factors. To date, this approach has been applied to MarA and SoxS in *E. coli* and *S.* Typhimurium [7,9,33]. For the latter organism, Rob and RamA have also been investigated [9]. Since this approach measures binding of the regulator to DNA, pleiotropic changes to gene expression are not a confounding factor. Furthermore, even if a given marbox can be shared by multiple regulators, this does not prevent detection of occupied sites. This strategy revealed that *E. coli* MarA stimulates expression of i) exonuclease VII, important for surviving treatment with DNA gyrase inhibitors ii) a regulatory cascade that ultimately represses biofilm formation and iii) the *mlaFEDCB* operon that controls lipid trafficking to reduce permeability of the cell outer membrane [7,9,12]. A drawback of the ChIP-seq approach is that it fails to identify all targets for a given regulator [34]. Most likely, this is because some DNA targets are low affinity or crosslink with bound proteins poorly. In other cases, the epitope needed for

immunoprecipitation may be obscured [35]. Anecdotally, our own ChIP-seq analyses suggest that MarA-like proteins elicit comparatively weak signals in ChIP-seq assays [7,9,36–40]. Thus, we identified only ~20% of known MarA targets in *E. coli*.

In this work, we have re-examined the set of *E. coli* genes controlled by MarA. We show that bioinformatic searches can identify genuine marboxes if combined with other genome-scale analyses. Specifically, if putative sites are filtered, based on location and orientation, with respect to other chromosomal features, many false positive predications are removed. Furthermore, false negatives are reduced, with ~70% of known MarA targets recovered. In this way, and by application of biochemical and genetic tools, we have identified new targets for MarA and SoxS involved in lipopolysaccharide (LPS) synthesis and cell wall remodelling. We show that regulation of these processes acts synergistically with control of the *mlaFEDCB* operon, that controls outer membrane integrity [41]. Thus, cells lacking the ability to properly regulate outer membrane properties, along with either LPS or cell wall synthesis, become hypersensitive to antimicrobial treatment.

## Materials and methods

### Strains, plasmids and oligonucleotides

Strains, plasmids and oligonucleotides are listed in S1 Table. To construct pET28a derivatives encoding *soxS* or *rob* the respective open reading frames were amplified using PCR in such a way that the gene sequence was flanked by *Nde*I and *Bam*HI restriction sites for subsequent digestion and then ligation with pET28a vector. Standard approaches for bacterial culture and nucleic acid isolation were used throughout.

### Bioinformatics

To search the *E. coli* genome for potential marboxes we used the search pattern function of the Colibri database [42]. The same tool was used to identify those predicted sites within the first 200 base pairs upstream of a gene start codon. The sequences of predicted sites were aligned to the *E. coli* genome (accession number U00096.2) using Bowtie 2 and then compared to our previous list of 28,107 *E. coli* transcription start sites (TSSs) [43,44]. The position and orientation of each putative marbox, with respect to the closest transcription start site, was determined using the logic functions in Microsoft Excel. Briefly, we allowed marboxes in any orientation centred between promoter positions -35 and +25, relative to the transcription start position. Class I marboxes were those in the reverse orientation, relative to the associated promoter, between positions -56 and -85. Forward orientation binding sites between promoter positions -36 and -55 were designated Class II. For predicted binding sites, correctly positioned with respect to a transcription start site, the function of the adjacent gene was recovered from the PANTHER database [45]. Genes were then clustered into functional groups manually. As a control, each *E. coli* gene was assigned a different number using the Microsoft Excel random number function. The genes were then sorted, according to the assigned number, and the first 200 genes selected. These arbitrarily selected genes were then allocated to the functional groups linked to the presence of a marbox.

### Proteins

Core RNA polymerase was purchased from NEB. The $\sigma^{70}$ subunit of the enzyme was purified as described previously [46]. The RNA polymerase holoenzyme was generated by incubating core enzyme with a 4-fold excess of $\sigma^{70}$ factor at room temperature for 20 minutes prior to use. To purify MarA we used the protocol of Kettles *et al* [12]. For

overexpression of SoxS or Rob, the appropriate pET28a derivatives were first used to transform T7 express cells. Transformants were cultured in LB medium and protein expression was induced with 0.4 mM IPTG. Cells were harvested by centrifugation at 1,600 x g for 10 minutes at 4 ºC and resuspended in 40 ml of 50 mM Tris-HCl (pH 7.5), 1 mM EDTA and 1 M NaCl. Cells were disrupted using an Avestin Emulsiflex C3 high pressure motorised homogeniser and inclusion bodies collected by centrifugation at 57,000 x g for 30 minutes. Pelleted material was resuspended in 50 mM Tris-HCl (pH 8.5) with 4 M urea. Particulates were again collected by sonication and finally resuspended in 50 mM Tris-HCl (pH 8.5) with 6 M guanidinium-HCl. Any remaining insolubilised material was removed by centrifugation and the supernatant was passed over a 1 ml Ni-sepharose HisTrap column. The column was washed with 50 mM Tris-HCl (pH 8.5) and 1 M NaCl before bound protein was eluted with an imidazole gradient. Fractions containing the overexpressed protein were identified by SDS-PAGE and then pooled before dialysis against 1 M NaCl, 50 mM HEPES, 1 mM dithiothreitol, 5 mM EDTA and 0.1 mM Triton X-100 overnight. Protein samples were concentrated to 1 mg/ml using a 5000 molecular weight cut-off Vivaspin 20 column. His tags were removed by adding 15 µl of thrombin-sephaose beads per mg of protein and incubating at room temperature for 5 hours with gentle rocking. Thrombin beads were removed by centrifugation of 1,600 x g for 5 minutes. Digested His tags were removed using the HisTrap column with flow-through retained and dialysed against 1 M NaCl, 50 mM HEPES, 1 mM dithiothreitol, 5 mM EDTA, 0.1 mM Triton X-100 and 20% (*v/v*) glycerol overnight. Recombinant proteins were stored at -20 ºC.

## Electrophoretic mobility shift assays

DNA fragments for use in electrophoretic mobility shift assays (EMSA) were prepared by PCR amplification, digestion with *Eco*RI, and purification by gel extraction. Fragments were radio-labelled at the 5′ end by addition of 1 µl T4 polynucleotide kinase and 1 µl [γ-32P]-ATP in manufacturer provided reaction buffer. Unincorporated nucleotides were removed by passing labelled DNA through two G-50 Sephadex columns. Electrophoretic mobility shift assays were done as described previously [47]. Samples were separated by PAGE at 150 V for 1-2 hours. Resulting gels were vacuum dried, exposed to a phosphorscreen overnight, then visualised using a Biorad FX phosphorimager.

## Primer extension

Primer extension was done to map transcription start sites as described by Haycocks and Grainger [48]. Briefly, RNA was extracted from JCB387 cells harbouring pRW50 containing the *lpxC* regulatory DNA fragment. Extractions were done using a Qiagen RNeasy mini kit according to the manufacturer's instructions and residual DNA was removed using a Turbo DNA-free kit (Ambion). RNA integrity was then checked by visualisation following agarose gel electrophoresis. The ratio of absorbance at 260 nm and 280 nm was used to assess the purity of the RNA. Primer extension products were analysed on a 6% denaturing polyacrylamide gel alongside size standards generated by manual sequencing of M13 phage DNA.

## in vitro transcription assays

Experiments were done exactly as described in previous work [7,12]. Products of *in vitro* transcription were separated by SDS PAGE and dried gels were exposed to a phosphor screen overnight before being scanned with a Biorad FX phosphorimager. Protein concentrations used in each experiment are indicated in figure legends.

### β-galactosidase assays

Promoter DNA fragments of interest were ligated upstream of *lacZ* in plasmid pRW50. The *E. coli* strain JCB387 was transformed with these constructs and used for β-galactosidase assays. The strain also carried plasmid pJ203 or a derivative providing low level constitutive expression of *marA*. Assays were done in triplicate and error bars show the standard deviation of the mean.

### Cell permeability assays

Membrane permeability was measured using a crystal violet assay described by Halder *et al* [49]. Briefly, cell cultures were harvested at 4500 x g for 5 minutes at 4 ºC. Cells were washed three times and re-suspended in 0.5 mM phosphate-buffered saline (PBS) (pH 7.4) containing 10 μg/ml crystal violet. This suspension was incubated at 37 ºC for 10 minutes and cells were then removed by centrifugation at 13,400 x g for 15 minutes. The absorbance of the supernatant at $OD_{590}$ was measured and the percentage uptake of crystal violet was calculated using the starting solution for comparison.

### Minimum inhibitory concentration determination

Overnight cultures were diluted to an $OD_{650}$ of 0.05 in sterile distilled water, before being further diluted 1 in 20 in LB. A 96-well flat bottomed microtiter plate was prepared using the double dilution method [50]. Each well contained 50 μl diluted culture in a final volume of 100 μl per well with an appropriate concentration of antibiotic. A growth control was included, with LB and culture only. To check media sterility, we also included wells with LB only. The plate was incubated for 18 hours at 37 ºC. The MIC was recorded as the lowest concentration that prevented bacterial growth. *E. coli* strains NCTC 10418 and ATCC 25922 were used as controls. Results were only accepted if the MIC was within one double dilution of the expected MIC for these strains.

### Genome editing using FRUIT

To introduce chromosomal point mutations, we used Flexible Recombineering Using Integration of *thyA* (FRUIT) [51]. To generate JCB387 Δ*thyA*, genomic DNA from MG1655 Δ*thyA* was isolated. The primers JW472 and JW473 were then used to amplify *thyA* flanking regions. The PCR product was purified using a Qiagen Gel Extraction kit and used to transform JCB387 by electroporation. Importantly, the JCB387 cells contained plasmid pACBSR, encoding the λ red genes under the control of an arabinose inducible promoter. Prior to electroporation, cells were grown in LB containing 0.2% arabinose. After electroporation, cells were recovered at 37 ºC for one hour and plated on M9 minimal medium containing 100 μg/ml thymine, 20 μg/ml trimethoprim and chloramphenicol. Desired colonies were identified using colony PCR with primers flanking the *thyA* site. Successful Δ*thyA* recombinants were streaked onto the same medium and stocks stored at -80 ºC until required. To create point mutations in the chromosome of JCB387 Δ*thyA* we first used PCR to amplify the *thyA* expression cassette from plasmid pAMD001. Each primer used for amplification had an 80 bp 5' sequence that matched the desired site of future mutation. PCR products were purified using a Qiagen Gel Extraction kit and used to transform electrocompetent JCB387 Δ*thyA* cells, containing pACBSR, which had previously been grown in LB containing 35 μg/ml chloramphenicol, 100 μg/ml thymine and 0.2% arabinose to induce expression of the λ red genes, by electroporation. After recovery for one hour at 37 ºC, cells were plated on M9 minimal medium (lacking thymine) containing chloramphenicol and incubated at 37 ºC for between 24 and 48 hours.

Desired colonies were identified using colony PCR with primers flanking the expected site of *thyA* insertion. Colony PCR products were then sequenced and successful *thyA+* intermediates were re-streaked onto the same medium. The desired chromosomal sequence, containing point mutations, was synthesised as a gene strand by Eurofins and amplified by PCR. Products were purified using a Qiagen Gel Extraction kit and used to transform the *thyA+* intermediate by electroporation. As in prior steps, the strain to be transformed carried pACBSR and was grown in the presence of arabinose, to induce expression of the λ red genes, before being made electrocompetent. Following electroporation, cells were recovered at 37 ºC for one hour, plated on M9 minimal medium containing 100 µg/ml thymine, 20 µg/ml trimethoprim and 35 µg/ml chloramphenicol, and incubated at 37 ºC for 24-48 hours. Desired colonies were identified using colony PCR with primers flanking the expected site of mutagenesis. Colony PCR products were then sequenced to confirm successful mutagenesis.

## Results

### Genome-wide identification of marbox elements correctly positioned for gene regulation

Previous bioinformatic screens for occurrences of the marbox identified ~10,000 instances across the *E. coli* chromosome [29,30]. The vast majority of these predicted sites are non-functional with respect to gene regulation [7,9]. In the intervening years, it has emerged that MarA-like factors often identify DNA targets via an unusual "pre-recruitment" mechanism [52–54]. Briefly, whilst most transcriptional activators bind their DNA target, and then recruit RNA polymerase, MarA-like proteins can bind RNA polymerase before promoter recognition [55]. This offers enhanced specificity by ensuring only functional marboxes, close to RNA polymerase binding elements, are recognised. Importantly, several studies have now mapped transcription start sites genome-wide in *E. coli* [44,56–58]. Hence, we incorporated this information into our bioinformatic screen outlined in Fig 1a. First, we interrogated the *E. coli* genome sequence for occurrences of the marbox, defined here as 5′-gcannwwntgnnaaa-3′, with a maximum of two mismatches. Note that, unlike many regulator binding sites, the marbox is not a palindrome and has a specific orientation. Hence, we scanned both DNA strands and identified 17,730 potential marboxes (Fig 1a, Step 1). Next, since most primary TSSs for mRNAs are close to a gene start codon, we excluded marboxes > 200 base pairs away from the 5′ end of known genes (Fig 1a, Step 2). We then calculated the distance between each of the remaining 2,051 marboxes and the nearest transcription start site. Importantly, this selection step also considered marbox orientation. Specifically, only marboxes in Class I or Class II positions, or overlapping the core promoter, were retained (Fig 1a, Step 3). Using these criteria, a further 1,056 marboxes were excluded (S2 Table). Whilst the selection steps above reduced the original number of predicted marboxes by >17-fold, ~70% of known MarA targets were retained (S3 Table) suggesting many of the sites could be functional.

### Clustering of identified marboxes according to adjacent gene function

In a final step, to identify marboxes likely to be used for gene regulation, we grouped sites according to adjacent gene function (Fig 1a, Step 4, and S2 Table). Our logic was that marbox occurrence, adjacent to genes with closely related roles, is unlikely to be coincidental. As a control, we also selected genes randomly and sorted them into the same functional categories. We then determined the percentage of marbox associated, compared to randomly selected, genes in each category (S1 Fig). Strikingly, marbox adjacent genes were enriched in 7 of the 16 functional groups. These encode factors involved in tRNA production, cell wall modulation,

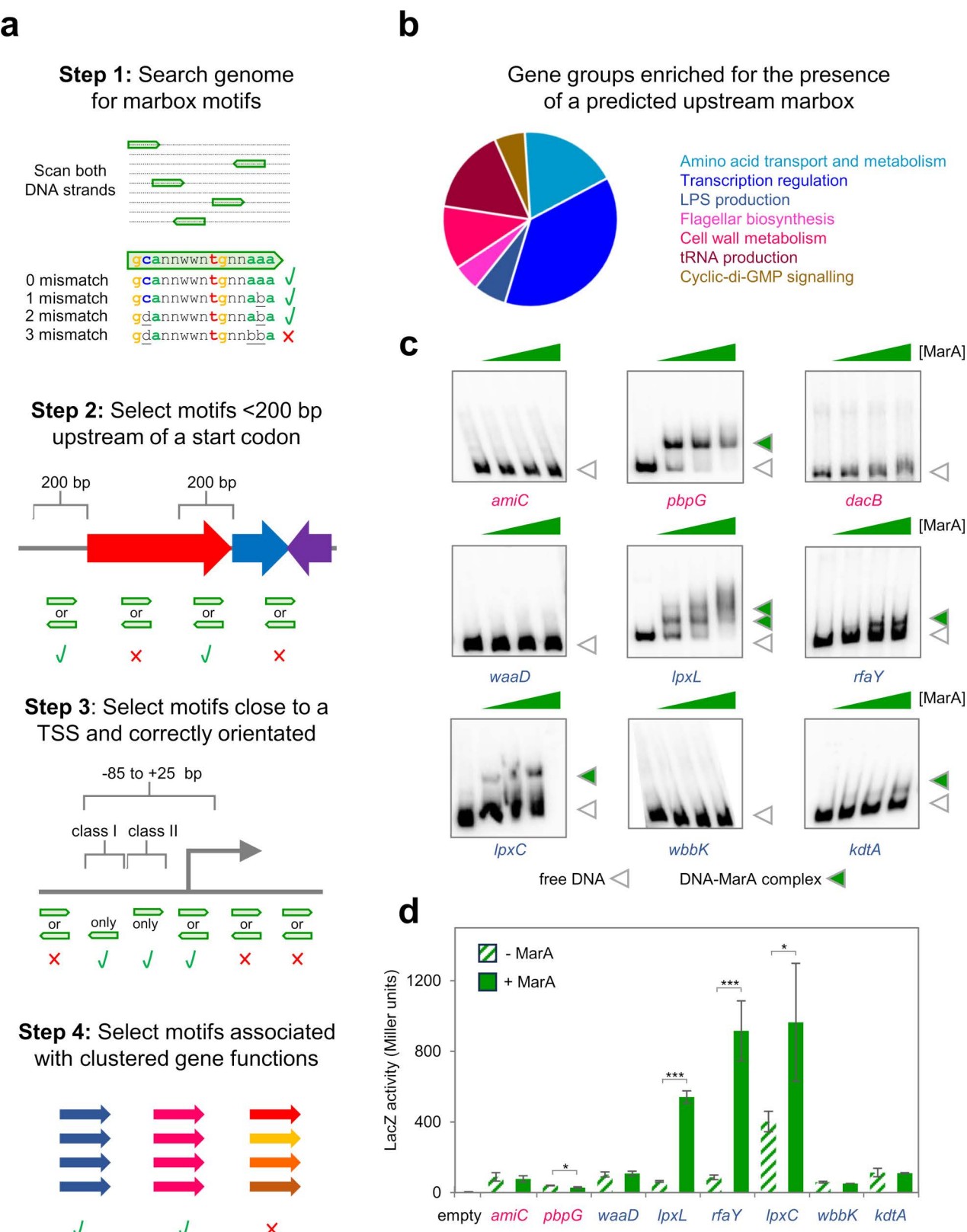

**Fig 1. Systematic scanning of the *Escherichia coli* chromosome for functional marbox motifs identifies regulation of LPS synthesis and cell wall remodelling. a. Strategy for identification of functional marboxes.** In Step 1, we searched the *E. coli* genome for DNA sequences with a

maximum of two mismatches to the consensus marbox, defined in our previous work [7]. In the schematic, the genome sequence is represented by a dashed line and marboxes are shown as headless green block arrows. The orientation of the arrow indicates the orientation of the marbox sequence. Note that sequences in the reverse orientation will match the consensus on the bottom, rather than top, DNA strand. Hypothetical examples of permissible and unacceptable marbox sequences are shown. Step 2 eliminated marboxes that were not within 200 base pairs of a gene start codon. Either marbox orientation was permitted, genes are shown as arrows, and the DNA is shown in grey. Step 3 excluded marboxes that were not appropriately positioned, and orientated, with respect to transcription start sites (TSSs) [44]. The bent arrow indicates a TSS and acceptable positions and orientations are marboxes are indicated by the schematic. Convention dictates that class I marboxes are in the reverse orientation whilst class II marboxes are in the forward orientation. In Step 4, we assigned functions of adjacent genes to each marbox. We then grouped the marboxes according to gene function. In the schematic, genes are shown as arrows and coloured according to function. Where multiple genes with related functions were identified, the associated marboxes were retained in our collection. **b. Overrepresented gene functions adjacent to marboxes selected by our analysis.** The pie chart indicates the number of genes in each group. **c. Binding of MarA to putative marboxes *in vitro*.** Each panel shows the result of an electrophoretic mobility shift assay with 1, 2 or 4 μM purified MarA protein. The DNA fragments correspond to DNA regions immediately upstream of start codons for indicated genes. The colour of gene name text is matched to the gene functions listed in panel b. Unbound DNA is indicated by an open triangle and MarA-DNA complexes are shown by equivalent green triangles. Uncropped gel images are in S6 Fig. **d. Regulation of gene expression by MarA at predicted target promoters.** The bar chart shows β-galactosidase activity measurements from cells without (stripes) or with (solid) ectopic MarA expression. In each case, except for the empty plasmid control, expression of *lacZ* is driven by the indicated promoter DNA fragment, cloned upstream of *lacZ*, in plasmid pRW50. Assays were done in triplicate on three separate occasions. Error bars show the standard deviation of the mean obtained from each set of triplicate experiments. Statistical significance was determined using a one-way ANOVA and post-hoc Tukey's HSD test. Significant differences between groups are indicated where P is < 0.05 (*) or < 0.001 (***).

flagellar production, LPS synthesis, amino acid metabolism, and cyclic-di-GMP signalling (Fig 1b). These account for 254 of the 17,730 marboxes originally identified.

## MarA binds to gene regulatory regions important for LPS synthesis and cell wall remodelling

To identify new MarA regulated genes, we chose a subset of potential targets from two of the gene functional groups. For genes involved in cell wall remodelling we chose *amiC* and *pbpG*. For functions related to LPS synthesis, we selected *lpxL*, *lpxC*, *rfaY*, and *kdtA*. As a control we included three regulatory regions not selected by our analysis. These were upstream of *dacB* (encoding a cell wall modification protein), *waaD* and *wbbK* (both encoding factors needed for LPS biosynthesis). Note that the *wbbK* regulatory region contains a putative marbox that was discarded in Step 3 of our selection. We tested binding of purified MarA protein to the different DNA regions *in vitro* using electrophoretic mobility shift assays (EMSAs). The results of the analysis are shown in Fig 1c. Whilst MarA did not bind to any of the control DNA fragments all but one (*amiC*) of the predicted targets was bound by MarA *in vitro*.

## MarA regulates expression of genes important for LPS synthesis and cell wall remodelling

Next, to determine if MarA was able to regulate transcription from the targets identified, the various DNA fragments were cloned upstream of *lacZ* in plasmid pRW50. As a control, we also cloned three of the promoter regions that did not bind MarA (*amiC*, *waaD* and *wbbK*). The resulting constructs were used to transform *E. coli* JCB387 cells containing plasmid pJ203, with or without *marA*. Briefly, this permits low level constitutive MarA production without the need to relieve repression of the chromosomal *marRAB* operon by MarR. The results of β-galactosidase assays are shown in Fig 1d. Changes in *lacZ* expression were not detected for any of the DNA fragments that did not interact with MarA *in vitro*. Conversely, changes in activity for all but one (*kdtA*) of the promoters, bound by MarA *in vitro*, were evident. In three cases, *lpxC*, *lpxL*, and *rfaY*, all involved in LPS biosynthesis, MarA activated gene expression. For *pbpG*, implicated in cell wall remodelling, transcription was repressed.

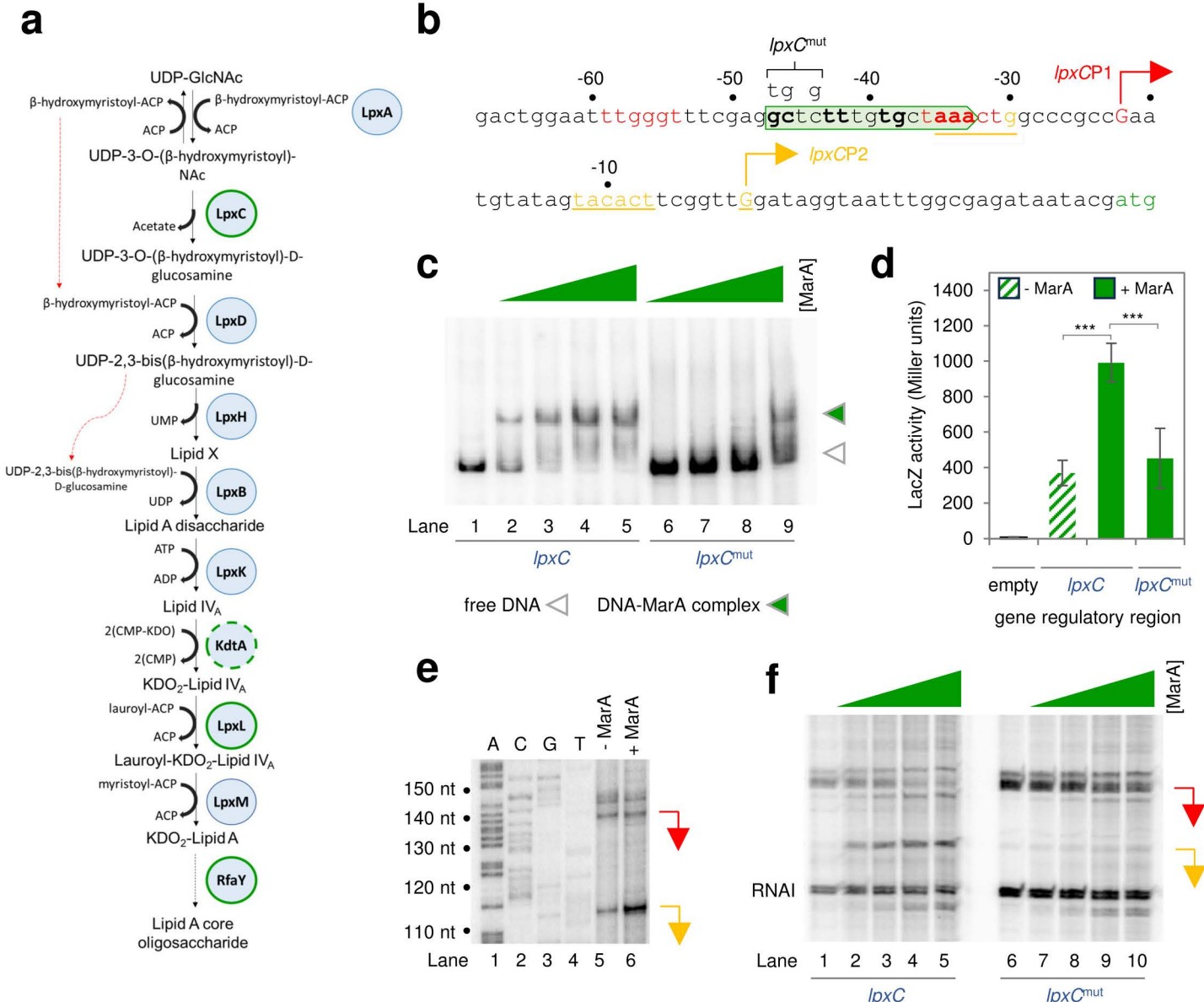

**Fig 2. *lpx*CP2 is a class II MarA dependent promoter. a. Schematic of the pathway for Lipid A-core oligosaccharide biosynthesis.** Enzymes are shown as circles and those whose expression is activated by MarA are outlined in green. Note that we detected binding of MarA to the *kdtA* regulatory but saw no regulatory effect in the conditions of our experiment. Reaction products are shown by text and reaction pathways by arrows. Solid arrows show enzymatic reactions, reactants and products. Dashed black arrows indicate biosynthetic steps that have been simplified. Red dashed arrows show recycling of reaction products. **b. Sequence of the *lpxC* regulatory region.** The *lpxC* start codon is green. Transcription start sites (TSSs) are indicated by bent arrows and labelled *lpxC*P1 and *lpxC*P2. The sequence of the intergenic region is numbered with respect to the *lpxC*P2 TSS. Promoter -10 and -35 elements are in coloured text and, for lpxCP2 only, also underlined (note the *lpxC*P2 -35 element overlaps the *lpxC*P1 -10 element). The marbox identified by our bioinformatic screen is boxed by a headless block arrow and bases in bold typeface are those matching the marbox consensus at key positions. Base changes made in the *lpxC*^mut derivative of the regulatory region appear above the starting DNA sequence and are labelled. **c. Binding of MarA to the *lpxC* regulatory region requires the predicted marbox.** Result of an electrophoretic mobility shift assay with the wild type or mutated *lpxC* regulatory region. Free DNA fragments are marked by an open triangle and MarA-DNA complexes by an equivalent green triangle. Where present, MarA was added at concentrations of 1, 2, or 4 μM. Uncropped gel images are in S6 Fig. **d. Upregulation of *lpxC* by MarA requires the predicted marbox.** The bar chart shows results of a β-galactosidase activity assay using lysates from cells carrying derivatives of the *lpxC* regulatory region, fused to *lacZ* in plasmid pRW50, in the presence (solid bars) or absence (striped bars) of ectopic MarA production.. Assays were done in triplicate on three separate occasions. Error bars show the standard deviation of the mean obtained from each set of triplicate experiments. Statistical significance was determined using a one-way ANOVA and post-hoc Tukey's HSD test. Significant differences between groups are indicated where P is < 0.001 (***). **e. MarA activates transcription from *lpxC*P2 *in vivo*.** The image shows the result of a primer extension assay to detect cDNAs, generated from *lpxC* mRNAs, transcribed from the regulatory region cloned in plasmid pRW50. Note that the gel is calibrated with sequencing reactions generated using an unrelated DNA template. The positions of the *lpxC*P1 and *lpxC*P2 transcription start sites

are indicated by red and orange bent arrows respectively. Uncropped gel images are in S6 Fig. **f. MarA activates transcription from *lpxC*P2 *in vitro* in a marbox dependent manner.** The image shows results of an *in vitro* transcription assay, using the *lpxC* regulatory region, or *lpxC*ᵐᵘᵗ derivative, cloned in plasmid pSR upstream of the λ*oop* terminator, as a DNA template. RNA polymerase σ⁷⁰ holoenzyme was present at a concentration of 4 μM and MarA was present at concentrations of 1, 2, 4 or 5 μM. The *lpxC*P1 and *lpxC*P2 derived transcripts are indicated by red and orange bent arrows respectively. Uncropped gel images are in S6 Fig.

### Expression of **lpxC** *is directly activated by MarA from a class II marbox*

LPS consists of O-specific polysaccharide, or O antigen, linked to the lipid A-core oligosaccharide. The biosynthetic pathway for the latter is shown in Fig 2a. Our data show that MarA activates expression of at least three of these enzymes (outlined in green in Fig 2a) and may also impact production of KdtA (dashed green outline on Fig 2a). The LpxC enzyme controls the first committed step in lipid A-core biosynthesis, hydrolysing UDP-3-O-(3-hydroxymyristoyl)-N-acetylglucosamine to UDP-3-O-(3-hydroxymyristoyl)glucosamine and acetate. The *lpxC* regulatory region, shown in Fig 2b, contains two promoters and a single predicted marbox in the forward orientation. First, to confirm correct identification of the marbox, we introduced mutations (labelled *lpxC*ᵐᵘᵗ in Fig 2b) and repeated our EMSA experiment. As expected, mutation of the marbox greatly reduced MarA binding (Fig 2c). Similarly, this also prevented activation by MarA *in vivo*, as judged using β-galactosidase assays (Fig 2d). Organisation of the *lpxC* regulatory DNA is curious, since the marbox overlaps the *lpxC*P1 core promoter elements whilst being positioned correctly for class II activation of *lpxC*P2 (Fig 2b). To understand how MarA might differently impact the two promoters *in vivo* we used primer extension assays. Briefly, RNA was purified from JCB387 cells harbouring the *lpxC* regulatory region, cloned in plasmid pRW50, and pJ203 with or without *marA*. We then used reverse transcriptase to generate a cDNA from a radiolabelled primer annealed to the pRW50 derived *lpxC* transcript. As expected, we detected RNAs originating from both *lpxC*P1 and *lpxC*P2 (Fig 2e, lane 5). Note that two transcripts, slightly different in size, originate from *lpxC*P1, likely because there is flexibility with respect to the base used for transcription initiation. Ectopic expression of MarA specifically stimulates transcription from *lpxC*P2 whilst having little impact of *lpxC*P1 (compare lanes 5 and 6). In a final set of experiments, we used *in vitro* transcription assays to monitor RNAs generated from each promoter. In these experiments, the regulatory DNA fragment is cloned upstream of the λ*oop* transcriptional terminator in plasmid pSR. The resulting DNA construct is provided as a template for purified RNA polymerase in the presence and absence of MarA. The results are shown in Fig 2f, where the RNAI transcript arises from the pSR replication origin and serves as an internal control. In the absence of MarA, RNA species were generated from *lpxC*P1 only. Presumably, in our primer extension assays, low levels of chromosomally derived MarA are sufficient to stimulate basal transcription from *lpxC*P2 (compare Fig 2e lane 5 and Fig 2f lane 1). Addition of MarA resulted in activation of *lpxC*P2 (Fig 2f, lanes 2-5). This activation was abolished with the *lpxC*ᵐᵘᵗ DNA template (lanes 6-10).

### Expression of both **lpxL** *and* **rfaY** *is also activated by MarA from a class II marbox*

The enzyme LpxL acts later than LpxC in the LPS synthesis pathway, generating lauroyl-$KDO_2$-Lipid $IV_A$. The sequence of the regulatory region, containing two promoters, in shown in Fig 3a. Our bioinformatic analysis identified a potential marbox overlapping the *lpxL*P1 -35 element. We confirmed correct marbox identification by altering the putative MarA site (labelled *lpxL*ᵐᵘᵗ¹ in Fig 3a). This greatly reduced interaction with MarA in EMSAs (Fig 3b). Consistent with class II MarA dependent activation, marbox mutation prevented stimulation

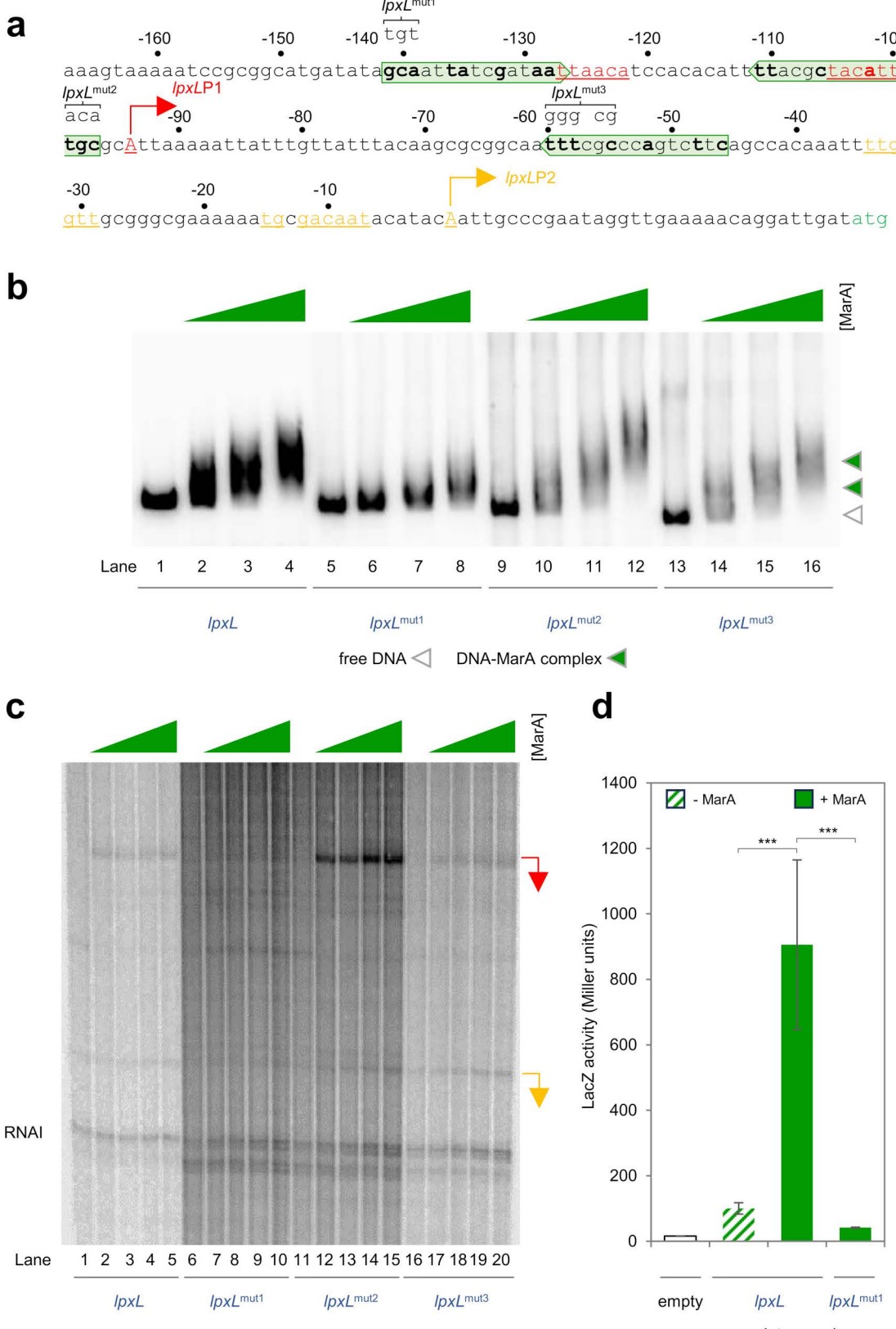

**Fig 3. *lpxL*P1 is a class II MarA dependent promoter. a. Sequence of the *lpxL* regulatory region.** The *lpxL* start codon is green. Transcription start sites (TSSs) are indicated by bent arrows and labelled *lpxL*P1 and *lpxL*P2. The sequence of the

intergenic region is numbered with respect to the *lpxL*P2 TSS. Promoter -10 and -35 elements are in coloured text and under-lined. The marboxes identified by our bioinformatic screen are boxed by a headless block arrows and bases in bold typeface are those matching the marbox consensus at key positions. Base changes made in the three different *lpxL*<sup>mut</sup> derivatives of the regulatory region, each with a different putative marbox mutated, appear above the starting DNA sequence and are labelled. **b. Binding of MarA to the *lpxL* regulatory region requires the predicted marbox overlapping the *lpxL*P1 -35 hexamer.** Result of an electrophoretic mobility shift assay with the wild type, or three different mutated, *lpxC* regulatory regions. Free DNA fragments are marked by an open triangle and MarA-DNA complexes by an equivalent green triangle. Where present, MarA was added at concentrations of 1, 2, or 4 μM. Uncropped gel images are in S6 Fig. **c. MarA activates transcription from *lpxL*P1 *in vitro* and requires the predicted marbox overlapping the *lpxL*P1 -35 hexamer.** The image shows results of an *in vitro* transcription assay, using the *lpxL* regulatory region, or different *lpxL*<sup>mut</sup> derivatives, cloned in plasmid pSR upstream of the λ*oop* terminator, as a DNA template. RNA polymerase σ<sup>70</sup> holoenzyme was present at a concentration of 4 μM and MarA was present at concentrations of 1, 2, 4 or 5 μM. The *lpxL*P1 and *lpxL*P2 derived transcripts are indicated by red and orange bent arrows respectively. Uncropped gel images are in S6 Fig. d. **Up regulation of *lpxL* by MarA *in vivo* requires the predicted marbox overlapping the *lpxL*P1 -35 hexamer.** The bar chart shows results of a β-galactosidase activity assay using lysates from cells carrying derivatives of the *lpxL* regulatory region, fused to *lacZ* in plasmid pRW50, in the presence (solid bars) or absence (striped bars) of ectopic MarA production. Assays were done in triplicate on three separate occasions. Error bars show the standard deviation of the mean obtained from each set of triplicate experiments. Statistical significance was determined using a one-way ANOVA and post-hoc Tukey's HSD test. Significant differences between groups are indicated where P is < 0.001 (***).

of *lpxL*P1 *in vitro* (Fig 3c, compare lanes 1-5 and 6-10) and rendered the regulatory region unresponsive to MarA *in vivo* (Fig 3d). Hence, MarA activates *lpxL*P1 by a class II mechanism. We note that the *lpxL*<sup>mut1</sup> fragment still exhibits some MarA binding activity in our EMSAs (Fig 3b, lanes 5-8). Similarly, the wild type regulatory region formed multiple complexes with MarA in these assays (Figs 3b, lanes 1-4, and 1c). Hence, the region may contain additional low affinity marboxes not found by our bioinformatic screen. We identified, and mutated, two potential sites (*lpxL*<sup>mut2</sup> and *lpxL*<sup>mut3</sup>, Fig 3a). The location and orientation of these predicted sites is suitable for class I activation of *lpxL*P2. However, the changes made had no impact on MarA binding or transcription (Fig 3b and 3c). We conclude that MarA regulates *lpxL* via promoter P1 only.

The final step in the LPS biosynthetic pathway is catalysed by the product of the *rfaY* gene. The upstream regulatory DNA resembles the *lpxL* intergenic region, with respect to promoter and marbox location (compare Figs 3a and 4a). Consistent with *rfaY*P1 also being a class II MarA-dependent promoter, mutation of the associated marbox prevents MarA binding, and activation of the promoter *in vitro* (Fig 4b and 4c). Similarly, mutation of the marbox renders *rfaY* expression unresponsive to MarA *in vivo* (Fig 4d).

### Expression of lpxC, lpxL *and* rfaY *can also be activated by SoxS and Rob*

As discussed above, SoxS and Rob often share DNA binding targets with MarA. Thus, we repeated our DNA binding and *in vitro* transcription assays to understand the roles of SoxS and Rob at the *lpxC*, *lpxL* and *rfaY* regulatory regions. As expected, both regulators bound to each of the regulatory DNA sections (S2a Fig). Note that, consistent with prior reports, Rob has a propensity to bind DNA non-specifically at higher concentrations *in vitro*. Consistent with direct binding, we also detected activation of transcription from *lpxC*P2, *lpxL*P1 and *rfaY*P1 by both SoxS and Rob *in vitro* (S2b Fig). In a final set of assays, with the three regulatory regions, we repeated our β-galactosidase expression analysis. Specifically, we measured *lacZ* expression from each regulatory region with or without ectopic production of SoxS. We did not test activity with expression of Rob since, in the absence of dipyridyl, Rob is sequestered in non-functional aggregates [22]. Further, inclusion of dipyridyl in the growth media is likely to induce activity of chromosomally encoded Rob. Consistent with our *in vitro* transcription analyses, expression of ectopic *soxS* stimulated all three regulatory DNA regions (S2c Fig). We note that Lee and co-workers also showed that SoxS could stimulate *rfaY* expression,

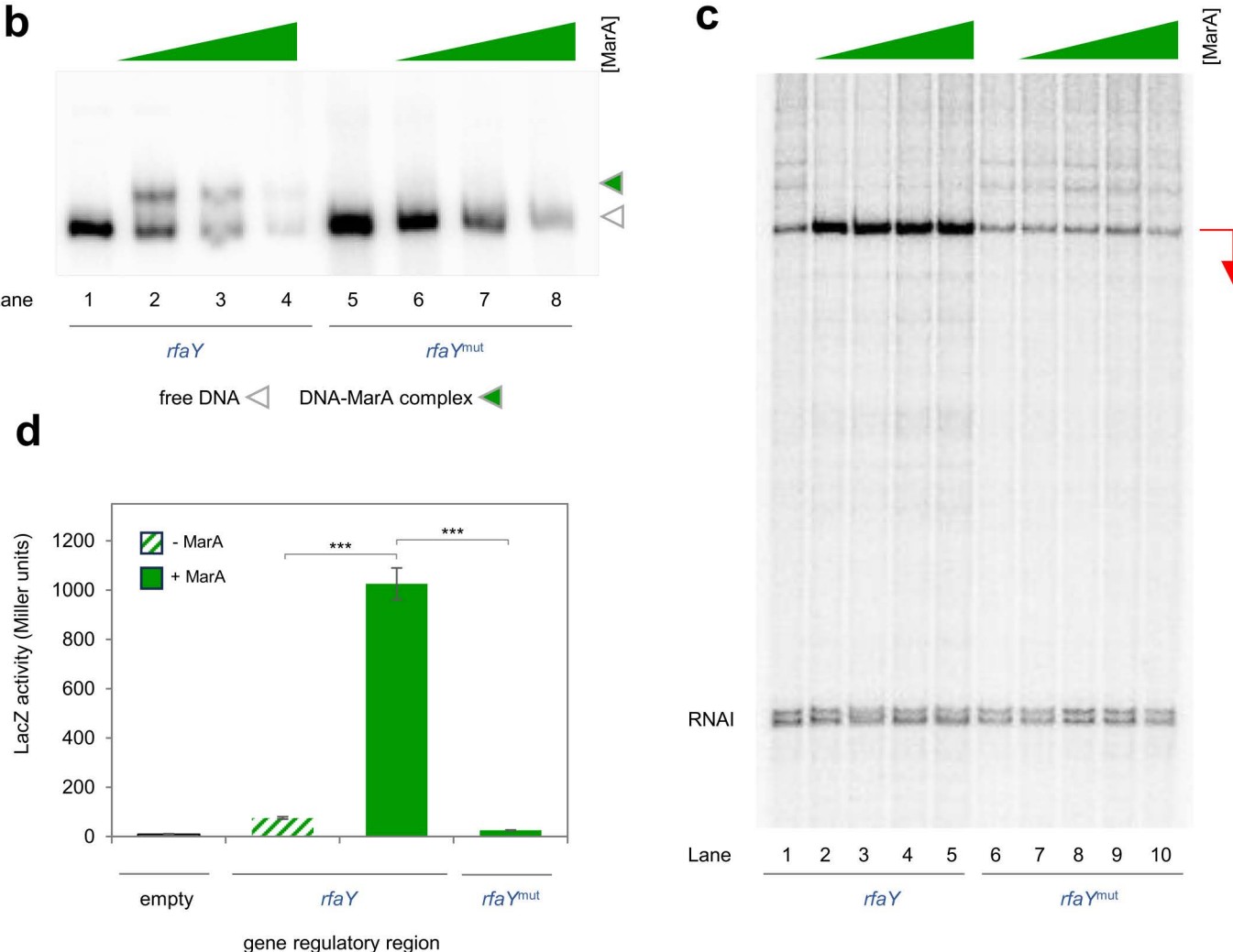

**Fig 4. *rfa*YP1 is a class II MarA dependent promoter. a. Sequence of the *rfaY* regulatory region.** The *rfaY* start codon is green. The transcription start site (TSS) is indicated by a bent arrow and labelled *rfa*YP1. The sequence of the intergenic region is numbered with respect to the *rfa*YP1 TSS. Promoter -10 and -35 elements are in coloured text and underlined. The putative marbox is boxed by a headless block arrow and bases in bold typeface are those matching the consensus at key positions. Base changes made in the *rfaY*mut derivative appear above the starting DNA sequence and are labelled. **b. Binding of MarA to the *rfaY* regulatory region requires the predicted marbox overlapping the *rfa*YP1 -35 hexamer.** Result of an electrophoretic mobility shift assay with the wild type or mutated *rfaY* regulatory regions. Free DNA fragments are marked by an open triangle and MarA-DNA complexes by an equivalent green triangle. Where present, MarA was added at concentrations of 1, 2, or 4 μM. Uncropped gel images are in S6 Fig. **c. MarA activates transcription from *rfa*YP1 *in vitro* and requires the predicted marbox.** The image shows results of an *in vitro* transcription assay, using the *rfaY* regulatory region, or *rfaY*mut derivative, cloned in plasmid pSR upstream of the λ*oop* terminator, as a DNA template. RNA polymerase σ70 holoenzyme was present at

a concentration of 4 μM and MarA was present at concentrations of 1, 2, 4 or 5 μM. The *rfaY*P1 transcript is indicated by red bent arrow. Uncropped gel images are in S6 Fig. **d. Up regulation of *rfaY* by MarA *in vivo* requires the predicted marbox.** The bar chart shows results of a β-galactosidase activity assay using lysates from cells carrying derivatives of the *rfaY* regulatory region, fused to *lacZ* in plasmid pRW50, in the presence (solid bars) or absence (striped bars) of ectopic MarA production.. Assays were done in triplicate on three separate occasions. Error bars show the standard deviation of the mean obtained from each set of triplicate experiments. Statistical significance was determined using a one-way ANOVA and post-hoc Tukey's HSD test. Significant differences between groups are indicated where P is < 0.001 (***).

in response to oxidative stress, and that this simulation was lost if DNA sequence upstream of the *rfaY*P1 -35 element was deleted [59].

### Expression of pbpG *is directly repressed by MarA, SoxS and Rob*

The three regulatory regions we identified as MarA targets, upstream of genes involved in LPS biosynthesis, have remarkably similar properties; all possess MarA independent activity and MarA activates by a class II mechanism in each case. The protein product of *pbpG* is not involved in LPS production. Instead, PbpG is a DD-endopeptidase that can hydrolyse D-alanyl-DAP amide bonds, within cell wall murein crosslinks [60]. The upstream regulatory DNA contains a single promoter and the predicted marbox overlaps the promoter spacer region (Fig 5a). Mutation of the marbox completely abolishes MarA binding in EMSAs (Fig 5b). Recall that ectopic expression of MarA repressed transcription from *pbpG*P in β-galactosidase assays (Fig 1d). Surprisingly, we were unable to recapitulate this observation *in vitro* with purified MarA (Fig 5c, lanes 1-5). However, transcription from *pbpG*P was completely repressed by SoxS in equivalent assays (lanes 6-10). Whilst we have no definitive explanation for this anomaly, our prior work demonstrated that the preference of a DNA site for MarA or SoxS *in vitro* is sensitive to buffer conditions [61]. Hence, inside the cell, it is possible that both regulators can utilise the site. Consistent with this, repression due to expression of SoxS *in vivo* was similar to that caused by MarA (Fig 5d). We conclude that MarA-like regulators oppositely control production of enzymes needed for LPS production (Figs 2-4) and cell wall turnover (Fig 5).

### lpxC, lpxL, rfaY *and* pbpG *are subject to simultaneous regulation by salicylic acid*

The gene targets for MarA involved in LPS biosynthesis are also regulated by SoxS and Rob (S2 Fig). However, whilst MarA and SoxS reduce *pbpG* expression *in vivo*, only SoxS has a dramatic impact on *pbpG* expression *in vitro* (Fig 5). Given the potential for differential regulation of the various targets, we wanted to understand if environmental conditions, known to specifically trigger the *mar* regulon, could simultaneously modify transcription of all four genes. To test this, we measured gene expression from each regulatory region in response to treatment of cells with salicylic acid, a phenolic inducer of *marRAB*. The results of the analysis are shown in S3 Fig. Addition of salicylic acid activated expression of *lpxC*, *lpxL* and *rfaY*. Simultaneously, expression of *pbpG* was abolished.

### Loss of genes needed for LPS production and cell wall turnover have opposing effects on resistance to tetracyclines

We next wanted to determine how the targets for MarA identified above might impact antibiotic resistance. As a first step, we used available phenotypic profiling data [62]. In the Fig 6a heatmap, each row represents an *E. coli* strain lacking the indicated gene. We also included data for *mlaE*, part of the *mlaFEDCB* operon, which controls outer membrane permeability,

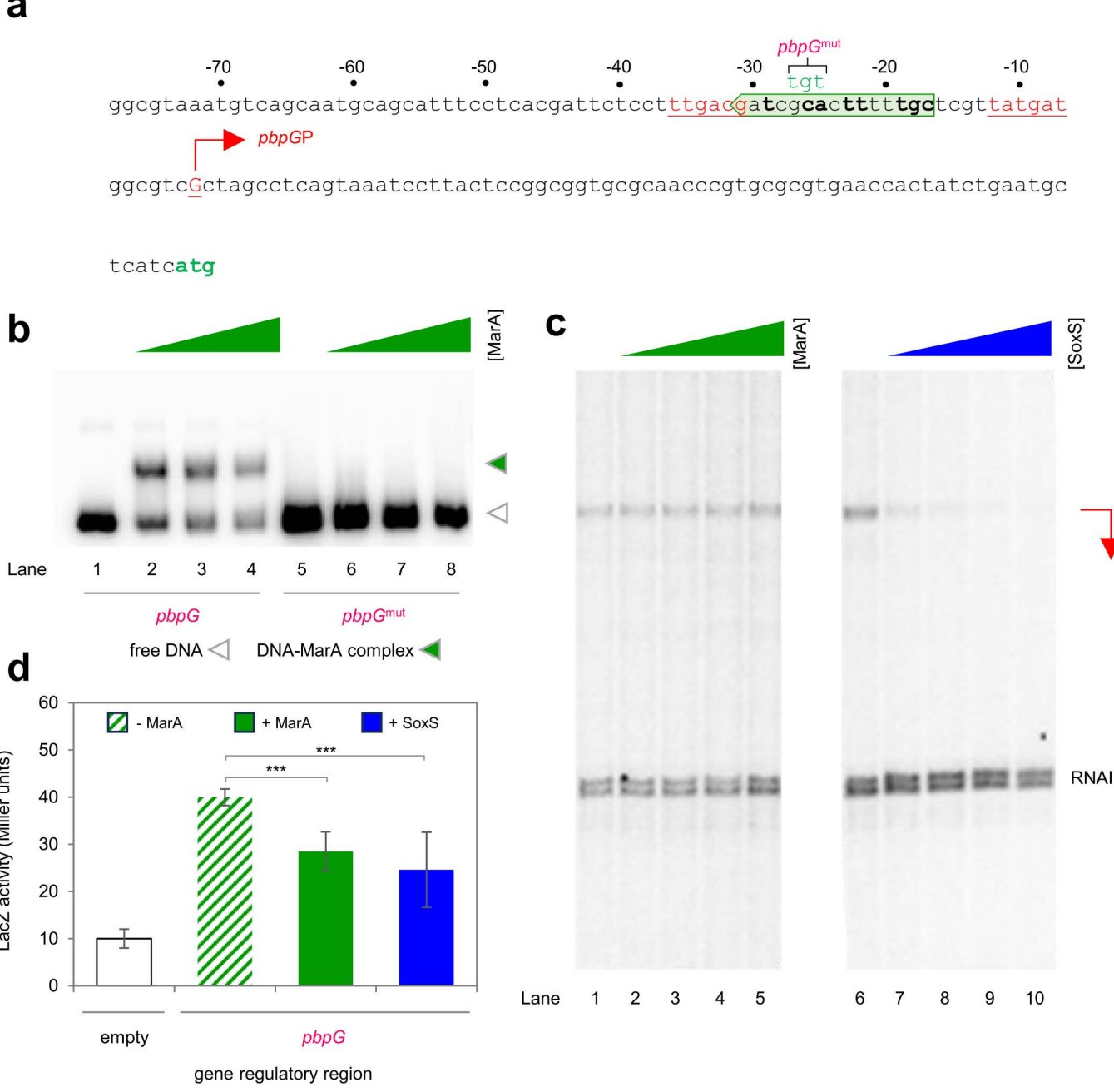

**Fig 5. MarA and SoxS are direct repressors of *pbpG* expression. a. Sequence of the *pbpG* regulatory region.** The *pbpG* start codon is green and the *pbpG* transcription start site (TSS) is indicated by a bent arrow *pbpG*P. The sequence of the intergenic region is numbered with respect to the TSS and promoter -10 and -35 elements are in coloured text and underlined. The putative marbox is boxed by a headless block arrow and bases in bold typeface are those matching the consensus at key positions. Base changes made in the *pbpG*mut derivative appear above the starting DNA sequence and are labelled. **b. Binding of MarA to the *pbpG* regulatory region requires the predicted marbox.** Result of an electrophoretic mobility shift assay with the wild type or mutated *pbpG* regulatory region. Free DNA fragments are marked by an open triangle and MarA-DNA complexes by an equivalent green triangle. Where present, MarA was added at concentrations of 1, 2, or 4 μM. Uncropped gel images are in S6 Fig. **c. MarA and SoxS represses transcription from *pbpG*P *in vitro* and require the predicted marbox.** The image shows results of an *in vitro* transcription assay, using the *pbpG* regulatory region cloned in plasmid pSR upstream of the λ*oop* terminator, as a DNA template. RNA polymerase σ70 holoenzyme was present at a concentration of 4 μM. Both MarA and SoxS were used at final concentrations of 1, 2, 4 or 5 μM. The *pbpG*P transcript is indicated by red bent arrow. Uncropped gel images are in S6 Fig. **d. Down regulation of *pbpG* by MarA and SoxS *in vivo*.** The bar chart shows results of a β-galactosidase activity assay using lysates from cells carrying the *pbpG* regulatory region, fused to *lacZ* in plasmid pRW50, in the presence (solid bars) or absence (striped bars) of ectopic MarA (green) or SoxS (blue) production.. Assays were done in triplicate on three separate occasions. Error bars show the standard deviation of the mean obtained from each set of triplicate experiments. Statistical significance was determined using a one-way ANOVA and post-hoc Tukey's HSD test. Significant differences between groups are indicated where P is < 0.001 (***).

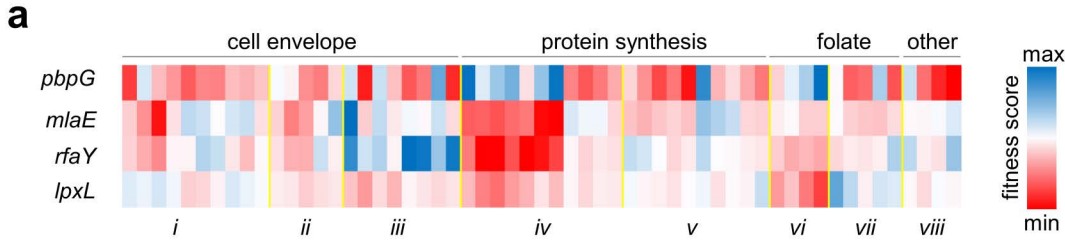

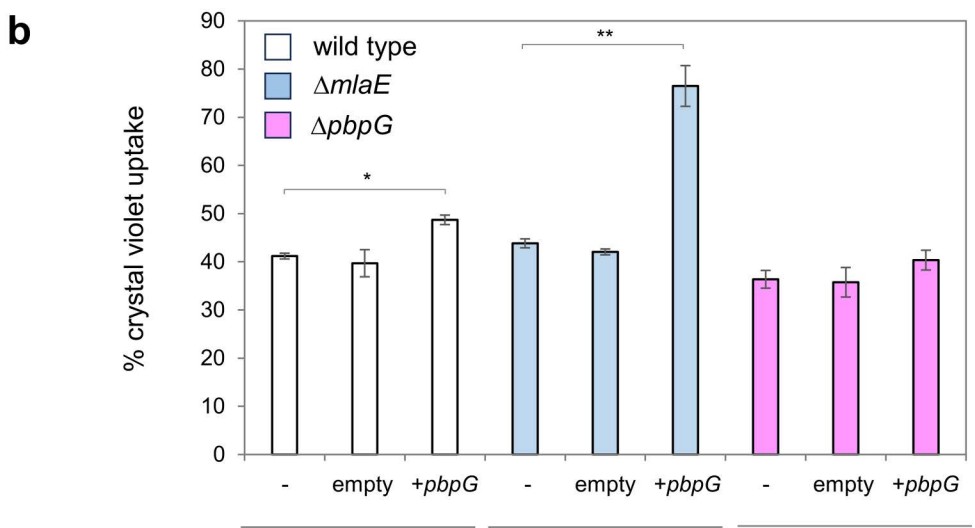

**Fig 6. Opposing expression of *pbpG* and genes for cell surface modification potentiate intrinsic antibiotic resistance. a. Impact of deleting MarA target genes on intrinsic antibiotic resistance.** The heatmap illustrates fitness scores of strains lacking MarA target genes (y-axis) compared to the wild-type parent strain. Strains were grown in the presence of different antibiotics (x-axis). The antibiotics are clustered according to the cellular process targeted (labelled above heatmap). Drugs are further divided into classes *i* through *viii* by yellow dotted lines. The classes are as follows: i cephalosporins, ii glycopeptides, iii cyclic peptides, iv tetracyclines; v aminoglycosides, vi antifolates vii sulfonamides and viii miscellaneous. The complete set of drugs tested, and their concentrations, are shown in S4 Fig. **b. Constitutive expression of *pbpG* is deleterious for cell envelope barrier function in the absence of *mlaFEDCB* expression.** The bar chart shows percentage crystal violet uptake by wild type, Δ*mlaE* or Δ*pbpG* cells. Uptake for each strain was also measured after transformation with empty pJ203 or a derivative for constitutive expression of *pbpG*. Results shown are means from three independent experiments and error bars represent standard deviation. Statistical significance was determined using a two-tailed student's t-test assuming unequal sample variance. Where indicated **P** = <0.05 (*) or <0.01 (**).

and is activated by MarA, for comparison [7]. Note that *lpxC* is essential so cannot be tested in this way. Each column represents a different antibiotic and rows are clustered according to the process targeted by each antibiotic. For each process, antibiotics are further grouped according to the antimicrobial class to which they belong (labelled *i* through *viii* in Fig 6a). Where rows and columns intersect, cells are coloured according to fitness score, a measure of colony growth on agar plates containing each of the various antibiotics [62]. Cells lacking *pbpG* have increased fitness in the presence of tetracyclines (Fig 6a, group iv) whilst the inverse is true for cells lacking *mlaE*, *rfaY* or *lpxL*. This inverse relationship is striking given the opposing regulatory effects of MarA.

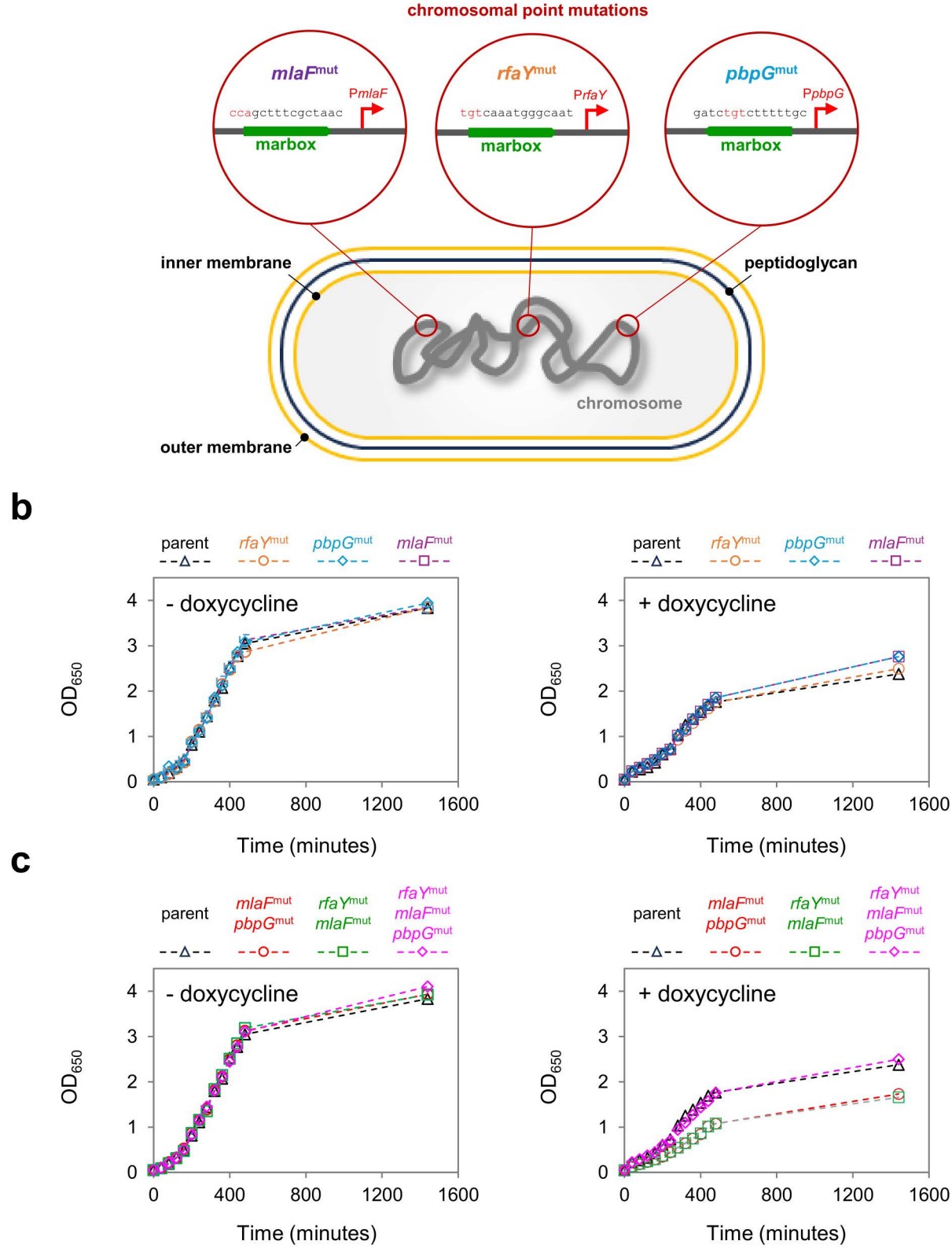

**Fig 7. Rewiring of the MarA regulon reveals synergy between regulatory targets. a. Schematic representation of MarA regulon rewiring.** The cartoon shows an *E. coli* cell. The chromosome is shown as a solid grey line, the inner and outer membranes are yellow, and the cell

wall is in blue. The expansions depict the three marboxes altered by point mutations (red text). The three sets of mutations were made individually or in three different combinations (*mlaF*<sup>mut</sup> + *pbpG*<sup>mut</sup>, *rfaY*<sup>mut</sup> + *mlaF*<sup>mut</sup>, and *rfaY*<sup>mut</sup> + *mlaF*<sup>mut</sup> + *pbpG*<sup>mut</sup>). **b. Growth of strains with individual marbox mutations in the presence and absence of doxycycline.** The line graphs show $OD_{600}$ values for cultures of the indicated strain derivatives, grown in LB medium, in the presence or absence of 1 µg/ml doxycycline. The parent strain is JCB387 Δ*thyA*. All strains carried pJ203 constitutively expressing *marA*. Assays were done in triplicate and repeated on 3 separate days. Doubling times for each strain are shown in S5 Fig. **b. Growth of strains with multiple marbox mutations in the presence and absence of doxycycline.** See information for panel b. Doubling times for each strain are shown in S5 Fig.

## The mlaFEDCB operon *and* pbpG *have opposing but synergistic cell permeability effects*

Previously, we showed that MarA activates expression of the *mlaFEDCB* operon to improve barrier function of the cell outer-membrane, reducing uptake of antibiotics. Endopeptidases, such as *pbpG*, hydrolyse amide bonds that link peptidoglycan strands in cell walls [60]. Reduced endopeptidase activity will increase cross linking and cell wall density [63]. Hence, porosity of the cell wall will be reduced. We predicted that simultaneous loss of *mlaE*, and constitutive expression of *pbpG*, could create highly permeable cells. To test this, we used the crystal violet dye uptake assay of Halder *et al* [49]. The results are shown in Fig 6b. Individually, loss of *mlaE*, or constitutive expression of *pbpG*, promoted only a small increase in dye uptake. However, when combined, the two genetic differences synergised to create a much more substantial increase in dye acquisition.

## Regulon rewiring shows that opposing regulation of lpxC and pbpG is synergistically impact antibiotic resistance

Permeability of the cell envelope is important for intrinsic antibiotic resistance [28]. Production of LPS is a known determinant for cell permeability [64]. We reasoned that simultaneous activation of the *mlaFEDCB* pathway, and LPS biosynthesis genes, by MarA, could synergistically increase resistance to tetracyclines and other antibiotics. Similarly, given our observations above, for *pbpG* and *mlaFEDCB*, opposing regulation of these genes could have a similar impact. To test this, we made point mutations to the *E. coli* chromosome that inactivated the marboxes upstream of *pbpG*, *rfaY* or *mlaFEDCB* (we were unable to make a strain with point mutations in the *lpxL* regulatory region). A schematic representation of the different mutations is shown in Fig 7a. Note that all strains lacked *thyA*, to allow for selection, and counter-selection, during strain construction. In either the absence of presence doxycycline, the strains all grew similarly (Figs 7b and S5a). Next, we made pairwise combinations of the *mlaFEDCB* marbox mutation with the marbox upstream of either *pbpG* or *rfaY*. We also made a strain with mutations in all three marboxes. As observed for individual mutations, all grew normally in the absence of doxycycline (Figs 7c and S5b). In the presence of doxycycline, we observed a substantial growth defect when mutations in the *mlaFEDCB* marbox were combined with loss of marbox function for control of *pbpG* or *rfaY* (Figs 7c and S5b). Surprisingly, normal growth was observed when all three marboxes were mutated. We speculate that this could be due to the action of other regulators, compensatory expression of other factors, or suppressor mutations elsewhere in the chromosome. Next, we turned our attention to understanding the impact of marbox mutations on the minimum inhibitory concentration (MIC) of different antibiotics. The results for each strain are shown in Table 1. As expected, the individual marbox mutations had little impact. Conversely, pairwise combinations of mutations in the *mlaFEDCB* marbox, with those upstream of *pbpG* or *rfaY*, greatly reduced the MIC for doxycycline, kanamycin, and tetracycline. Combining all marbox mutations again had no impact.

**Table 1. Impact of marbox mutations in the *waaY*, *mlaF* and *pbpG* regulatory regions on antibiotic minimum inhibitory concentrations.**

| Minimum inhibitory concentration (MIC) (µg/ml)[2] | | | |
|---|---|---|---|
| Strain[1] | Doxycycline | Kanamycin | Tetracycline |
| Δ*thyA* | 8 – 32 | 2 – 16 | 4 – 16 |
| Δ*thyA waaY*[mut] | 8 – 16 (8) | 8 – 16 (8) | 8 – 16 (8) |
| Δ*thyA mlaF*[mut] | 8 – 16 (8) | 8 – 16 (16) | 4 – 8 (4) |
| Δ*thyA pbpG*[mut] | 16 | 16 | 8 |
| Δ*thyA pbpG*[mut] *mlaF*[mut] | 1 – 2 (2) | 2 | 2 |
| Δ*thyA waaY*[mut] *mlaF*[mut] | 1 – 4 | 1 – 2 (1) | 1 – 2 (2) |
| Δ*thyA pbpG*[mut] *waaY*[mut] *mlaF*[mut] | 8 – 16 (16) | 8 – 64 | 4 – 16 |

[1]All strains are derivatives of *E. coli* JCB387 and constitutively express *marA* from plasmid pJ203.

[2]Numbers shown are ranges and modal values are in parentheses.

## Discussion

Bacterial cells have evolved intrinsic stress response systems that can be used as a defence against antimicrobial compounds. Mutations that deregulate such systems can cause clinical levels of antibiotic resistance [28]. Hence, understanding underlying mechanisms can identify opportunities for intervention. In this work, we identify new regulatory targets for MarA, impacting different aspects of *E. coli* cell envelope biology, and show that these systems can act together to potentiate survival of treatment with antibiotics (Fig 8). Specifically, we previously showed that MarA activates expression of the *mlaFEDCB* operon, which encodes a lipid trafficking system. Loss of this activity results in accumulation of phospholipids in the outer leaflet of the outer membrane. As a consequence, barrier function of the cell envelope is compromised. Incorrect regulation of the *mlaFEDCB* operon synergises with marbox mutations impacting LPS production or cell wall remodelling (Fig 7). Synergy likely arises because increased LPS production, or reduced cell wall porosity, can compensate for an inability to upregulate phospholipid removal from the outer leaflet. If such compensatory effects are lost, more substantial defects in cell envelope barrier function are expected.

The properties of LPS are known to influence outer membrane stability and barrier function [65]. Strikingly, we identified several genes, all in the lipid A-core oligosaccharide biosynthesis pathway, subject to control by MarA. In each case, the gene regulatory logic is similar; MarA-independent transcription arises from a downstream promoter (Figs 2 and 3) or basal activity of the MarA targeted promoter (Fig 4). Most likely, this allows genes to be transcribed at levels important for general housekeeping. We suggest that, when repression by MarR is released, by any of the mechanisms discussed above, this expression is augmented by MarA. This can involve switching on an ancillary promoter (Figs 2 and 3) or upregulating basal promoter activity (Fig 4). We note that, in the absence of MarA, the *lpxC* regulatory region drives far higher levels of β-galactosidase expression than the equivalent DNA regions for *lpxL* or *rfaY* (Fig 1d). Conversely, when MarA is expressed, the gene control sequences are similarly active (Fig 1d). Hence, MarA has a bigger regulatory impact on *lpxL* and *rfaY* (activation by 9-fold and 13-fold respectively) compared to *lpxC* (a 2-fold stimulation) (Fig 1d). This is intriguing, as LpxC catalyses the first committed step in lipid A-core oligosaccharide biosynthesis (Fig 2a). Conversely, LpxL and RfaY control two of the three final steps in the pathway. We speculate that, without MarA, levels of LpxL and RfaY are restricted, compared to those of LpxC, limiting flux of intermediates

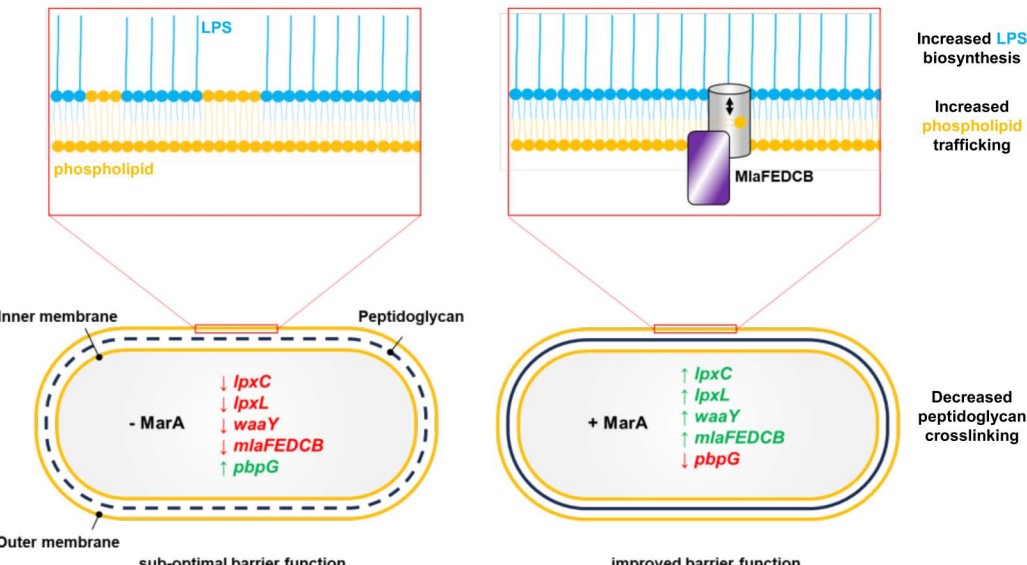

**Fig 8. Model for coordination of cell envelope biology by MarA.** The schematic shows changes to cell envelope biology coordinated by MarA. When MarA is expressed, genes important for lipopolysaccharide (LPS) biosynthesis (*lpxC*, *lpxL* and *rfaY*) and trafficking of phospholipids between the inner and outer leaflet of the outer membrane (the *mlaFEDCB* operon) are upregulated. This stimulates greater decoration of the cell surface with LPS and removal of phospholipids from the outer leaflet of the outer membrane, both of which improve barrier function [7,64]. Simultaneously, MarA represses expression of *pbpG*, that encodes an endopeptidase involved in cell wall hydrolysis. Loss of endopeptidase activity can lead to increased density and thickening of the cell wall [63]. This also contributes to better cell envelope barrier function.

through the system. When MarA is expressed, a surge in expression of LpxL and RfaY could relieve this metabolic bottleneck and permit rapid generation of lipid A-core oligosaccharide. We suggest that MarA-like regulators could act similarly in other bacteria. For example, in *K. pneumoniae*, RamA activates expression of LpxC and LpxL-2 [66]. We note that these authors also report much smaller stimulation of *lpxC* expression by RamA than is recorded for *lpxL-2*. Previous work also detected a signal for SoxS binding upstream of *lpxC* using chromatin immunoprecipitation, and reported upregulation of *lpxC* in response to oxidative stress [33]. Although these authors did not map the precise location of SoxS binding, it's likely that SoxS acts via the marbox described here.

To our knowledge, there are no previous reports of MarA-like regulators controlling expression of enzymes used for cell wall remodelling. However, it is intriguing that OxyR, a transcription factor which, like SoxS, controls the response to oxidate stress, activates *mepM*, encoding an alternative endopeptidase [33]. We speculate that repression of cell wall remodelling factors causes peptidoglycan changes that improve cell wall barrier function. Alone, this has little effect on sensitivity to antibiotics (Figs 6 and 7). However, in concert with other cell envelope changes, more drastic differences are apparent (Figs 6 and 7). Hence, loss of MarA mediated *pbpG* control, combined to mutation of the *mlaFEDCB* marbox, synergistically increases sensitivity to several antibiotics (Table 1). Taken together, these observations suggest that modulating cell wall properties can be co-ordinated with other aspects of cell envelope biology to reduce uptake of antibiotics. We suggest that MarA-like regulators are likely to impact further aspects of cell envelope biology, in ways we are yet to appreciate.

## Supporting information

**S1 Fig. Gene functions over represented adjacent to predicted marboxes.** The bar chart shows the distribution of genes between different functional categories. Two groups of genes are represented: those selected randomly (blue) and those adjacent to a predicted marbox (orange).
(PDF)

**S2 Fig. Regulation of *lpxL*, *lpxC* and *rfaY* by SoxS and Rob. a. Binding of SoxS and Rob to the lpxC, lpxL, and rfaY regulatory regions.** Results of electrophoretic mobility shift assays with the wild type derivatives of the three regulatory regions. Free DNA fragments are marked by an open triangle and SoxS-DNA or Rob-DNA complexes are highlighted by blue or red triangles respectively. Where present, SoxS was added at concentrations of 1, 2, or 4 μM. Conversely, Rob was used at concentrations of 0.2, 0.4 or 0.8 μM. Uncropped gel images are in S6 Fig. **b. Control of *lpxC*, *lpxL*, and *rfaY* transcription by SoxS and Rob.** The image shows results of *in vitro* transcription assays, using the indicated regulatory regions cloned in plasmid pSR upstream of the λ*oop* terminator, as a DNA template. RNA polymerase σ[70] holoenzyme was present at a concentration of 4 μM and SoxS was present at concentrations of 1, 2, or 4 μM. For Rob, concentrations of 0.2, 0.4 or 0.8 μM were used. The *lpxC* or *lpxL* P1 and P2 derived transcripts are indicated by red and orange bent arrows respectively. The *rfaY*P1 transcript is indicated by a bent red arrow. Uncropped gel images are in S6 Fig. **c. Activation of *lpxC*, *lpxL* and *rfaY* transcription by SoxS *in vivo*.** The bar chart shows results of a β-galactosidase activity assay using lysates from cells carrying the indicated regulatory regions, fused to *lacZ* in plasmid pRW50, in the presence (solid bars) or absence (striped bars) of ectopic SoxS production. Assays were done in triplicate on three separate occasions. Error bars show the standard deviation of the mean obtained from each set of triplicate experiments. Statistical significance was determined using a one-way ANOVA and post-hoc Tukey's HSD test. Significant differences between groups are indicated where P is < 0.05 (*) or < 0.001 (**).
(PDF)

**S3 Fig. Salicylic acid simultaneously alters expression of *lpxL*, *lpxC*, *rfaY* and *pbpG*.** The bar chart shows results of a β-galactosidase activity assay using lysates from cells carrying the indicated regulatory regions, fused to *lacZ* in plasmid pRW50, in the absence (open bars) or presence (green bars) of 5 mM sodium salicylate. Error bars show the standard deviation of three biological repeat experiments. Statistical significance was determined using a student's T-test. Significant differences between groups are indicated where P is < 0.05 (*), < 0.001 (**) or < 0.0001 (***).
(PDF)

**S4 Fig. Detailed heatmap view.** The figure shows a more detailed expansion of the heatmap in Figure 6b.
(PDF)

**S5 Fig. Doubling times for strains with marbox mutations in the presence and absence of doxycycline. a. Doubling times for individual marbox mutations.** The bar chart shows doubling times for strains carrying mutations in individual marboxes compared to the Δ*thyA* parent. Results are the average of three independent experiments and error bars indicate standard deviation. Statistical significance was determined using a two-tailed student's t-test assuming unequal sample variance. **b. Doubling times for combinations of marbox mutations.** As for panel a, except that strains carry combinations of marbox mutations. Statistical significance

was determined using a two-tailed student's t-test assuming unequal sample variance. Where indicated (\*\*) P = <0.01.
(PDF)

**S6 Fig. Uncropped gel images.** Regions of the gel images used in the main figures are boxed and labelled accordingly.
(PDF)

**S1 Table. Strains, plasmids and oligonucleotides.**
(DOCX)

**S2 Table. Predicted marbox sequence and position relative to the nearest transcription start site.**
(DOCX)

**S3 Table. Identification of known MarA target genes.**
(DOCX)

**S1 Data. Numerical data: The raw numerical data for graphs is provided and the precise figure these data are linked to is indicated in each case.**
(XLSX)

## Acknowledgements

We thank Joseph Wade for critically reading the manuscript prior to submission and providing advice on the application of FRUIT.

## Author contributions

**Conceptualization:** David C Grainger.

**Data curation:** Alexandra E. Trigg.

**Formal analysis:** Alexandra E. Trigg.

**Funding acquisition:** David C Grainger.

**Investigation:** Alexandra E. Trigg, David C Grainger.

**Methodology:** David C Grainger.

**Project administration:** David C Grainger.

**Resources:** David C Grainger.

**Supervision:** Prateek Sharma, David C Grainger.

**Visualization:** Alexandra E. Trigg.

**Writing – original draft:** David C Grainger.

**Writing – review & editing:** Alexandra E. Trigg, Prateek Sharma, David C Grainger.

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
