## [Decision Letter · Decision Letter 0]

5 Nov 2024

PGENETICS-D-24-01128Coordination of cell envelope biology by Escherichia coli MarA protein potentiates intrinsic antibiotic resistancePLOS Genetics Dear Dr. Grainger, Thank you for submitting your manuscript to PLOS Genetics. After careful consideration, we feel that it has merit but does not fully meet PLOS Genetics's publication criteria as it currently stands. Therefore, we invite you to submit a revised version of the manuscript that addresses the points raised during the review process. Please respond to all the points raised by the reviewers, in particular: - provide statistical analysis/quantification of the data where possible, and consider whether some of the statements/conclusions should be revised. - Make sure all figures are properly annotated/organized and that the sections in the manuscript are coherently organized and that the significance of the findings are discussed properly. - Describe and discuss the computational approach and selection of target genes in more detail. Please submit your revised manuscript within 60 days Jan 04 2025 11:59PM. If you will need more time than this to complete your revisions, please reply to this message or contact the journal office at plosgenetics@plos.org. Please include the following items when submitting your revised manuscript:* A rebuttal letter that responds to each point raised by the editor and reviewer(s). You should upload this letter as a separate file labeled 'Response to Reviewers '. This file does not need to include responses to any formatting updates and technical items listed in the 'Journal Requirements' section below.* A marked-up copy of your manuscript that highlights changes made to the original version. You should upload this as a separate file labeled 'Revised Manuscript with Track Changes '.* An unmarked version of your revised paper without tracked changes. You should upload this as a separate file labeled 'Manuscript '. If you would like to make changes to your financial disclosure, competing interests statement, or data availability statement, please make these updates within the submission form at the time of resubmission. Guidelines for resubmitting your figure files are available below the reviewer comments at the end of this letter. We look forward to receiving your revised manuscript. Kind regards, Morten KjosAcademic EditorPLOS Genetics Sean CrossonSection EditorPLOS Genetics

Aimée Dudley

Editor-in-Chief

PLOS Genetics

Anne Goriely

Editor-in-Chief

PLOS Genetics

**Journal Requirements:** **Additional Editor Comments (if provided):****Reviewers' comments:** Reviewer's Responses to Questions

**Comments to the Authors:**

Reviewer #1: The manuscript submitted by Trigg et al. describes a detailed analysis of selected members of the MarA regulon in E. coli and how they synergistically play a role in contributing to intrinsic antimicrobial resistance. The study is elegant and thorough, combining clever bioinformatic identification of likely MarA binding sites followed up with convincing molecular studies to verify the targets. I have no major concerns, but highlight several points below that, if clarified, would improve the quality of this manuscript:

• The background information cited on MarRA, SoxS and Rob can be confusing in parts. As these factors and their regulated genes are intrinsic, what are the natural signals for activating MarA (or even SoxS/Rob) regulation or MarR derepression? Salicylic acid is mentioned but how is this relevant? Is MarR derepression or activation of MarA regulated genes ever triggered by exposure to drugs? Can you elaborate on what stressful conditions (line 447) might be in reality? This would be useful for the reader to contextualise the otherwise beautifully presented data.

• The bioinformatic pipeline is clever but somewhat arbitrary. I agree that their choice of targets to follow up based on Figure S1 was logical but is there a possibility that interesting MarA targets may be missed? Is there any published transcriptome data of marA mutant cells that could be integrated here to assess other potentially regulated genes?

• How do the authors reconcile the lack of in vitro effect in figure 5c? I was not clear on what they concluded was actually going on here. Is it possible that both MarA and SoxS are needed in tandem? If they are added in combination to the assay in figure 5d do you get an exaggerated effect?

• Effect of pbpG in Figure S3 is difficult to discern. Are these statistically significant changes?

• Figure 6a – it would be helpful if you provide a key for the colour scheme on the heatmap

• Line 382 – Should be spelled “murein”

• Mechanistically how does PbpG and LpxC synergy work? I am not clear on what the authors are proposing here? Are they suggesting that these independent systems could somehow interact or that the synergy is purely at the regulatory level? It seems counterintuitive that its not a cumulative fitness defect and that they are entirely co-dependent? Perhaps I am missing something here that could be clarified in the discussion.

• Figure 7a – I don’t think the cell schematic is necessary. The images of the mutations is sufficient.

• Figure 7c – can you calculate some statistical analyses on the growth rates to quantify the extent of the fitness defects?

• Line 405-406 - What could the compensatory factors be? SoxS or Rob? Have you tried these assays in Rob/SoxS mutant backgrounds?

• Line 467 – Should be spelled “oxidative”

Reviewer #2: The manuscript of Trigg et al., describes a study in to the multi-antibiotic resistance activator proteins MarA that has been implicated in intrinsic resistance to antibiotic in enteric bacteria. The authors used a computational screen to determine the functional MarA binding sites based on proximity to know transcriptional start sites in the E coli genome. The authors then used genetic and biochemistry to interrogate the activation of potential targets such as genes affecting LPS biogenesis and PG remodelling. Uncoupling of both these processes from MarA regulation results in increased sensitivity to envelope stress and makes cells hypersensitive to mutations in the lipid membrane shuttling system, Mla that transports lipids between the two membranes in didermic bacteria.

The work is well conducted and makes a novel contribution to our understanding not only of intrinsic resistance but also in terms of interplay between envelope remodelling and antibiotics. This is particularly important given that MarA overexpression can allow acrAB and tolC mutants to become resistant to certain antibiotics and yet understanding this mechanism has been elusive to date due to the difficulty in identifying bona fide MarA binding sites.

I think the work is suitable for publication as it is, but i have a few minor points for the authors to consider

I wonder if the authors saw any MarA binding upstream of the Mce domain containing proteins that potentially have similar function to MlaD such as PqiAB or YebST? and or doe these mutations give the same phenotype?

Figure 7 B & C - growth curves should be on a log scale or semi-log plots

Figure S5 amiC and wbbK are not very convincing shifts...

Reviewer #3: The study has an exciting goal of using a combination of computational and experimental techniques to identify novel MarA binding sites. It provides an interesting computational approach and succeeds in finding novel MarA binding sites for several genes related to LPS and cell wall remodeling using a variety of experimental techniques.

Since the computational approach is a basis for the detailed experimental investigations, I wish it is described and evaluated in more details. Why were the sequences of the predicted sites aligned to the MG1655 genome when the study is done with JCB387? Do the targets identified with this approach overlap with the ones identified by other approaches? How were the targets for further investigations selected? Did they stand out statistically or they are chosen completely randomly?

The experimental results for identifying the marboxes and their classes are interesting, but the figures could be improved to enhance the clarity. Many figures could benefit from better annotations in figure instead of putting explanations in the figure legend. The gel images should be clearly labeled with DNA/protein size bars on the side, and the heatmaps need to have a colorbar legend to indicate the values the colors represent. Also, there are several places in the text where the readers were directed to compare parts of a specific panel to another part from a panel from a different figure. The writers should consider organizing the figures better so it is easier for the readers to understand the results.

There are 12 subsections of the results section. Each section is short and discusses different things. While they are rife with information about experimental design and presentation of the results, they do not offer detailed biological interpretation of the results. Any theories on why the in vitro assay did not show the influence of MarA on the expression of pbpG while the in vivo assay did? Any hypothesis on why cells lacking pbpG have increased fitness when exposed to tetracyclines, a drug that interferes primarily with protein synthesis, but show mixed patterns for drugs affecting cell envelope, which pbpG is directly associated with? The connections between some of the 12 sub-sections and how they contribute to the overall conclusion of the study is not obvious. The results appear scattered instead of coming together as a holistic and convincing story. It would be great to see some of these results reorganized and condense to have a simpler structure with more detailed biological discussion of what the results actually mean in the biological context.

**Have all data underlying the figures and results presented in the manuscript been provided?**

Reviewer #1: Yes

Reviewer #2: Yes

Reviewer #3: None

PLOS authors have the option to publish the peer review history of their article (what does this mean? ). If published, this will include your full peer review and any attached files.

**Do you want your identity to be public for this peer review?** For information about this choice, including consent withdrawal, please see our Privacy Policy .

Reviewer #1: No

Reviewer #2: No

Reviewer #3: No

---

## [Editor Report · Decision Letter 1]

26 Feb 2025

Dear Dr Grainger,

We are pleased to inform you that your manuscript entitled "Coordination of cell envelope biology by Escherichia coli MarA protein potentiates intrinsic antibiotic resistance" has been editorially accepted for publication in PLOS Genetics. Congratulations!

Yours sincerely,

Morten Kjos

Academic Editor

PLOS Genetics

Sean Crosson

Section Editor

PLOS Genetics

Aimée Dudley

Editor-in-Chief

PLOS Genetics

Anne Goriely

Editor-in-Chief

PLOS Genetics

Comments from the reviewers (if applicable):

**Data Deposition**

http://datadryad.org/submit?journalID=pgenetics&manu=PGENETICS-D-24-01128R1

**Press Queries**

---

## [Editor Report · Acceptance letter]

PGENETICS-D-24-01128R1

Coordination of cell envelope biology by Escherichia coli MarA protein potentiates intrinsic antibiotic resistance

Dear Dr Grainger,

We are pleased to inform you that your manuscript entitled "Coordination of cell envelope biology by Escherichia coli MarA protein potentiates intrinsic antibiotic resistance" has been formally accepted for publication in PLOS Genetics! Your manuscript is now with our production department and you will be notified of the publication date in due course.

With kind regards,

Zsofia Freund

PLOS Genetics

On behalf of:
